# Hybrid 3D-Printed and Electrospun Scaffolds Loaded with Dexamethasone for Soft Tissue Applications

**DOI:** 10.3390/pharmaceutics15102478

**Published:** 2023-10-17

**Authors:** Silvia Pisani, Valeria Mauri, Erika Negrello, Valeria Friuli, Ida Genta, Rossella Dorati, Giovanna Bruni, Stefania Marconi, Ferdinando Auricchio, Andrea Pietrabissa, Marco Benazzo, Bice Conti

**Affiliations:** 1Department of Drug Sciences, University of Pavia, Viale Taramelli, 12, 27100 Pavia, Italy; valeria.friuli@unipv.it (V.F.); ida.genta@unipv.it (I.G.); rossella.dorati@unipv.it (R.D.); bice.conti@unipv.it (B.C.); 2SC General Surgery 2, Fondazione IRCCS Policlinico San Matteo, Viale Camillo Golgi, 19, 27100 Pavia, Italy; v.mauri@smatteo.pv.it (V.M.); e.negrello@smatteo.pv.it (E.N.); a.pietrabissa@smatteo.pv.it (A.P.); 3Consorzio per lo Sviluppo dei Sistemi a Grande Interfase (C.S.G.I.), Department of Chemistry, Physical Chemistry Section, University of Pavia, 27100 Pavia, Italy; giovanna.bruni@unipv.it; 4Department of Civil Engineering and Architecture, University of Pavia, 27100 Pavia, Italy; stefania.marconi@unipv.it; 5Fondazione IRCCS Policlinico San Matteo, Viale Camillo Golgi, 19, 27100 Pavia, Italy; ferdinando.auricchio@unipv.it; 6Department of Clinical, Surgical, Diagnostic and Pediatric Sciences, University of Pavia, 27100 Pavia, Italy; m.benazzo@smatteo.pv.it; 7Department of Otorhinolaryngology, Fondazione IRCCS Policlinico San Matteo, 27100 Pavia, Italy; 8Integrated Unit of Experimental Surgery, Advanced Microsurgery and Regenerative Medicine, Università degli Studi di Pavia, 27100 Pavia, Italy

**Keywords:** electrospinning, 3D-printing, tissue engineering, dexamethasone, hybrid scaffolds

## Abstract

Background: To make the regenerative process more effective and efficient, tissue engineering (TE) strategies have been implemented. Three-dimensional scaffolds (electrospun or 3D-printed), due to their suitable designed architecture, offer the proper location of the position of cells, as well as cell adhesion and the deposition of the extracellular matrix. Moreover, the possibility to guarantee a concomitant release of drugs can promote tissue regeneration. Methods: A PLA/PCL copolymer was used for the manufacturing of electrospun and hybrid scaffolds (composed of a 3D-printed support coated with electrospun fibers). Dexamethasone was loaded as an anti-inflammatory drug into the electrospun fibers, and the drug release kinetics and scaffold biological behavior were evaluated. Results: The encapsulation efficiency (EE%) was higher than 80%. DXM embedding into the electrospun fibers resulted in a slowed drug release rate, and a slower release was seen in the hybrid scaffolds. The fibers maintained their nanometric dimensions (less than 800 nm) even after deposition on the 3D-printed supports. Cell adhesion and proliferation was favored in the DXM-loading hybrid scaffolds. Conclusions: The hybrid scaffolds that were developed in this study can be optimized as a versatile platform for soft tissue regeneration.

## 1. Introduction

After injuries, adjacent tissue cells, as well as progenitor cells that are recruited from the bone marrow, migrate and proliferate at the damaged site in order to rapidly restore the lost tissue. This process is often accompanied by a strong and long-lasting inflammatory response [1]. While a sudden inflammatory response is suitable for the defense against invading pathogens, it can limit the healing response and also cause the inappropriate activation of fibroblasts [2]. Fibroblasts are cells of mesenchymal origin that are found in almost every tissue, and are the main producers of the ECM in homeostatic conditions and in response to injury [3]. Their differentiation into myofibroblasts causes the deposition of large amounts of the extracellular matrix (ECM), called fibrosis, which prevents further regenerative processes [4].

To make the regenerative process more effective and efficient, tissue engineering (TE) strategies have been implemented. TE combines information from cell biology and materials science to mimic the physical and chemical properties of native tissue, with the aim of functional restoration following injury [5]. The fabrication of three-dimensional (3D) porous scaffolds allows for the achievement of supports that are able to sustain the survival, and guide the proliferation and functional differentiation, of cells, as well as the targeting of transplanted replacement or supporting cells. Scaffolds, due to their suitable designed architecture, offer the proper location of the position of cells, as well as cell adhesion and the deposition of the extracellular matrix (ECM). Moreover, scaffolds guarantee adequate gas transport, essential nutrients, and controlling factors to promote cell proliferation, survival, and differentiation [6]. The porosity that is guaranteed by 3D scaffolds must be optimized according to the specific cell line that is used and the task that they have to perform [7]. For example, it was reported that cell infiltration for dermal fibroblasts is enhanced by scaffolds with a 34.4% porosity and an average pore size of 100 μm [8]. In another work, it was reported that fibroblasts exhibit optimal cell proliferation in scaffolds with a pore size of 200–250 μm and a porosity of about 86% [9,10].

Depending on their material origin, scaffolds can be classified as either natural (made from, e.g., collagen, chitosan, glycosaminoglycans, hyaluronic acid, decellularized matrix, etc.), synthetic (made from bio-ceramics, metals, polyesters, etc.) or composite (made from a combination of materials of different origins) [11]. The important aspect is that all these materials are biocompatible (biomaterials) and do not create systemic toxicity (cytotoxicity, genotoxicity, mutagenicity, carcinogenicity, and immunogenicity) once implanted in the human body [12]. Bio-scaffolds should also have a tunable mechanical strength, a large surface area, and surface properties that mimic physical and ECM chemical properties in order to promote cell adhesion, proliferation, and differentiation [13,14]. In addition to cellular support, an advantage of three-dimensional scaffolds is their ability to also be used as systems for the delivery of drugs and/or active molecules (such as fibroblast growth factor) [15].

Among all types of scaffolds, electrospun nanofibers have attracted much attention for soft tissue regeneration due to their ease of construction and function modification to regulate their composition, morphology, structure, 3D architecture, and biological functions, as well as specific light/electric/magnetic properties [16]. Electrospinning is a versatile method for the production of nanofibers from various kinds of polymers. The diameter and morphology of electrospun nanofibers can be easily optimized by manipulating electrospinning parameters to fit with their final applications. Electrospinning parameters can be categorized into three main groups: (i) polymer solution (e.g., concentration, viscosity, surface tension, and conductivity), (ii) process (e.g., applied electrostatic potential, collection distance, and feed rate), and (iii) ambience (e.g., temperature, relative humidity (RH), and surrounding air velocity in the spinning chamber) [17,18]. Concerning the process parameters, several studies have found that higher voltages cause a reduction in the fiber diameter. Conversely, the pore size and fiber diameter increase when the polymeric solution flow rate increases. The type of collector exerts a significant influence on the alignment of electrospun fibers; a conductive metal plate is commonly used as a target to orient the fibers randomly in a nonwoven structure, while highly aligned fibers can be obtained by using a cylindrical drum. Finally, it was observed that the distance between the collector and the needle tip (cm) can also affect the final fiber diameter and morphology. If the distance is too short, the solvent cannot fully evaporate before it reaches the collector; on the contrary, a too-long distance can cause the formation of beaded fibers. An optimum distance should be set up to allow for the collection of defect-free and dry electrospun nanofibers [17,19].

Electrospun nanofibers demonstrate greater properties in terms of cell behavior manipulation, cell migration (providing topographical and biochemical cues), morphology (by regulating gene expression), and the targeted regulation of stem cell differentiation [20]. In fact, cell morphology is influenced by fiber morphology; cells exhibit a rounded morphology on nanofibers, randomly oriented fibers, and low-porosity scaffolds, while cells exhibit an elongated, spindle-shaped morphology on microfibers, aligned fibers, and high-porosity scaffolds. Cells migrate with higher velocities on nanofibers, aligned fibers, and high-porosity scaffolds, but they migrate greater distances on microfibers, random fibers, and highly porous scaffolds [21,22].

Besides cells and growth-controlling bioactives, electrospun scaffolds play a central role in TE that is analogous to the role performed by the extracellular matrix (ECM) in vivo. Mimicking the structure of a native ECM at the nanoscale level is one of the critical attributes demonstrated by nanofibers. This unique feature, as well as its customizable structure to befit different types of tissues, make nanofibers a competent candidate for tissue engineering [23]. For example, tendons, heart tissue, and blood vessels exhibit anisotropic arrangements with highly ordered structures, while random fibers can be used for tissues without a specific orientation, such as skin or adipose tissue. However, random fiber orientation is advantageous for tissue engineering, as it offers an increased surface area that promotes cell adhesion, proliferation, tissue regeneration, nutrient exchange, and drug release [24,25].

Electrospun nanofibers also represent a valid platform in TE for controlled drug delivery (DD). In fact, nanofibers’ high porosity and large surface area favor an increase in drug-loading efficiency and speed up the response of stimuli-delivered drugs [26]. In the material selecting phase, to achieve a typical drug release profile, the interaction between the drug and the fiber scaffolds should be considered. The composition, molecular weight, hydrophilicity, and degradation rate of the fiber polymer matrix affect drug release behavior. Moreover, the molecular mass, crystallinity, and solubility of the drug also affect the release behavior [27].

As far as the improvement of tissue regeneration and soft tissue healing is concerned, anti-inflammatory drugs have proven to be capable of collagen synthesis stimulation, and they increase the strength in the early phases of repair [28,29]. For example, dexamethasone accelerates muscle regeneration by modulating kinesin-1-mediated focal adhesion signals [30], and has also been applied to alleviate inflammation and reduce the loss of cartilage extracellular matrices [31]. Hydrocortisone has been used to improve the healing of chronic and burn wounds [32]. Metronidazole as been loaded into core/shell nanofibers to establish the minimal drug content that is necessary to achieve the appropriate anti-inflammatory effect [33]. Mirzaeei et al. designed and developed antibacterial/anti-inflammatory dual drug-loaded nanofibrous inserts for the ophthalmic sustained delivery of gentamicin and methylprednisolone [34].

In this work, dexamethasone (DXM), a steroidal anti-inflammatory drug, was embedded into biodegradable poly-L-lactide- co-poly- caprolactone (PLA/PCL 70:30) polymer scaffolds that were made using the electrospinning technique, which were combined with PLA/PCL 3D-printed scaffolds to obtain electrospun/3D-printed hybrid scaffolds with different polymer infills of 20% and 50%. The aim of this work was to test how the loaded DXM and the different scaffolds’ stiffness influenced cell growth and proliferation. The application of these scaffolds in carrying anti-inflammatory drugs is oriented towards soft tissue regeneration. 

## 2. Materials and Methods

### 2.1. Polymeric Scaffolds Preparation

The copolymer Poly-L-lactide-co–poly-caprolactone (PLA/PCL) at a 70:30 ratio (Resomer LC 703 S—Mw 160,000 Da, obtained from Evonik Industries, Evonik Nutrition & Care GmbH, 64275, Darmstadt, Germany), was used for the production of electrospun scaffolds (EL-S) obtained using the Nanon-01A electrospinning setup, equipped with a dehumidifier (MEEC instruments, MP, Pioltello, Italy). A 20% *w*/*v* polymeric solution was prepared in a solvent blend of methylene chloride (MC) and dimethylformamide (DMF) at a 70:30 ratio, respectively. As reported in previous work, MC was chosen because it is a good solvent for the PLA/PCL copolymer, and for its low boiling point (40 °C) that allows for fast solvent evaporation during the electrospinning process. DMF was selected for its high dielectric constant (36.70) that promotes fibers’ stretching when an electric field is applied [35]. Electrospinning process parameters, such as the voltage (30 kV), flow rate (1 mL/min), distance from needle–collector (15 cm), spinning time (40 min), 22 Gauge (G) needle (inner diameter: 0.759 mm), temperature (25 ± 3 °C), and relative humidity (30 ± 5%), were optimized in a previous study [35]. Placebo and dexamethasone (DXM, Mw = 392.46 g/mol, Sigma Aldrich, Milan, Italy)-loaded scaffolds were obtained, as reported in Table 1. The drug-loaded EL-S (#4) was obtained by dissolving the copolymer PLA/PCL (20% *w*/*v*) and DXM (0.04% *w*/*v*) in DCM/DMF at a 70:30 ratio. Both the placebo and the drug-loaded polymeric solutions were left under magnetic stirring (100 rpm) overnight.

The placebo and DXM-loaded polymeric solutions were also used to prepare hybrid scaffolds (HD-S). For the 3D-printed scaffolds’ (3DP-S) preparation, a PLA/PCL 70:30 copolymer filament (TreeD Filaments^®^ Company, Milan, Italy) was used. Fused filament fabrication (FFF) 3D-printing technology (LeapFrog HS^®^ (Dutch LeapFrog Group, Washington, DC, USA)) was employed for the scaffolds’ manufacturing. The 3DP-S were produced at different infill percentages (20% and 50%) according to a preliminary study performed by the authors [36]. The 3DP-S at 20% and 50% infill levels were placed on an electrospinning apparatus NANON01A collector and subsequently coated by electrospinning the PLA/PCL 20% *w*/*v* solutions (placebo and 0.04% *w*/*v* DXM-loaded) using the process parameters reported above.

### 2.2. Dexamethasone Quantification

The DXM-loaded scaffolds were cut (1.5 × 1.5 cm) and weighed. The drug/polymer weight ratio used for the preparation of the 10 mL solution (0.004/2 g = DXM 0.2% *w*/*w*) was taken into consideration for the theoretical calculation of DXM in the scaffolds. For the DXM quantification in the HD-S (20% and 50% infill), it was reported in previous studies that the weight of the electrospun fibers collected on the 3D-printed scaffold represents 20% of the total HS weight [36].

The scaffolds were solubilized in 1 mL of Thetrahydrofuran (THF, UV cutoff: 212–215 nm) and a DXM quantification was performed using UV spectrophotometric analysis [37]. The calibration curve (R^2^ = 0.9987) was plotted at increasing DXM concentrations (0.000315, 0.000625, 0.00125, 0.0025, 0.005, 0.01, 0.02, and 0.04 mg/mL) in THF. The absorbance was measured at 240 nm using quartz cuvettes (6705 UV/Vis Spectrophotometer, Single Mobile Holder, Jenway, Sungai Petani, Kedah, Malaysia). Placebo scaffolds were used as a reading control (blank).

The drug content % (Equation (1)) and encapsulation efficiency % (Equation (2)), were evaluated using the following equations:(1)$$Drug\;Content\;\left(\;\frac{w}{w}\%\right)=\frac{mg\;DXM\;in\;the\;scaffold}{mg\;scaffold}\times 100$$
(2)$$Encapsulation\;efficiency\;\%=\frac{\left(mg\right)\;DXM\;loaded\;into\;the\;scaffold\;}{\left(mg\right)\;DXM\;used\;to\;make\;the\;formulation\;\;}\times 100$$

### 2.3. Contact Angle Evaluation/Wettability

A Contact Angle Meter DMe-211 plus (KYOWA, J) with a CCD chamber was used to assess the scaffolds’ wettability. Each single scaffold was placed on the horizontal support of the equipment and a drop (9 ± 2 μL) of Hepes 0.1 M (Sodium N-2-Hydroxyethyl-piperazine-N′-2-ethansulfonate, Sigma Aldrich, Milan, Italy) was dropped onto the scaffold surface. The measurements were carried out at room temperature (25 ± 2 °C), and the liquid/scaffold contact time was fixed at 2 min. The software FAMAS automatically detected the volume of the drop delivered and the two characteristic parameters of the drop: the drop height (h) and the drop base (a). Using these parameters, the software automatically computed the contact angle (θ).

### 2.4. In Vitro Release Test in Static and Dynamic Conditions

In vitro release tests were performed in Hepes buffer 0.1 M (pH 7.2). A static release test was conducted using a semi-permeable membrane (dialysis membrane, standard RC tubing MWCO: 12–14 KD, Spectrum Laboratories Inc., Canada) for 7 days at 37 °C under magnetic stirring (300 rpm). A final volume of Hepes 0.1 M (20 mL) was set up to maintain sink conditions during all test phases [38]. At established timing points (1, 2, 4, 6, 24, 48, and 168 h), 1 mL of the medium was extracted from the dialysis external medium to be analyzed. 1 mL of fresh Hepes 0.1 M was added to reinstate the total volume. The samples were analyzed using the UV method for dexamethasone quantification. The calibration curve (R^2^ = 0.9995) was plotted at increasing DXM concentrations (0.000315, 0.000625, 0.00125, 0.0025, 0.005, 0.01, 0.02, and 0.04 mg/mL) in Hepes 0.1 M. The absorbance was measured at 240 nm using quartz cuvettes (6705 UV/Vis Spectrophotometer, Single Mobile Holder, Jenway, Sungai Petani, Kedah, Malaysia).

An in vitro release test was performed, also in dynamic conditions, for 7 days at 37 °C using an IVTech (IVTech Srl—Innovative Start up, Massarosa (LU), Italy) Livebox1 (LB1—tangential perfusion system) interconnected through a system of tubes to a peristaltic pump (IPC4—ISMATEC). The IVTech LiveBox 1 was filled up with 20 mL of Hepes 0.1 M, with a pH of 7.2, and the pump was activated to allow for a fluid flow rate of 0.4 mL/min. At established timing points (1, 2, 4, 6, 24, 48, and 168 h), 1 mL of the medium was extracted from the chamber to be analyzed and 1 mL of fresh Hepes 0.1 M was added to reinstate the total volume. The samples were analyzed using the UV spectrophotometric method (240 nm) for dexamethasone quantification in Hepes 0.1 M, pH 7.2.

Zero-order, first-order, Higuchi, Korsmeyer–Peppas and Hixson–Crowell kinetics models were used to define the drug release mechanisms of the loaded scaffolds (#4, #5, and #6) in the static and dynamic conditions tested. The highest correlation coefficient (R^2^) derived from the kinetics models will be used to describe the prevailing drug release mechanisms of the different samples [39].

### 2.5. Morphological Characterization

A scanning electron microscopy (SEM) analysis was conducted on the scaffolds to evaluate the following parameters: fiber diameter and porosity. The scaffolds were cut and put on an adhesive coal sample-holder and they were collocated on metal support. Before starting the analysis, the scaffolds were made conductive thanks to the deposition on their surface of a small layer of gold in an argon atmosphere to make interactions among the electrons possible. A Zeiss EVO MA10 apparatus (Carl Ziss, Oberkochen, Germany) was used. The analyses was conducted using 3.00 KX and 10.00 KX zooms. The resulting images were processed using ImageJ software, a digital image processing computer program that is supported by standard image processing functions that are able to calculate fibers’ diameter and porosity [40].

### 2.6. Comparative Mechanical Assessment

The purpose of this preliminary mechanical analysis was to check if the placebo samples had values that were in line with those calculated in the previous characterization [36]. The mechanical properties of the placebo and the DXM-loaded scaffolds were analyzed using a tensiometer (Mark-10 Tensile, MARK-TEN, Copiague, NY, USA) with a 25 N load cell. The scaffolds were cut in dog-bone shape, with a width equal to 0.4 cm and a length equal to 2.5 cm, using a die-cutting machine. This particular shape ensures analysis reproducibility, allowing the force to be applied at the same sample point (ISO 37:2017) [41]. The dog-bone-shaped scaffolds were subjected to an axial tensile test at room temperature (25 ± 3 °C) using a loading velocity of 5 mm/min, according to the literature, for soft tissue characterization [42,43]. The Young’s modulus (E) was computed as the slope of the stress–strain curve and compared with previously produced hybrid scaffolds [36]. The reliability of the E computation was assessed through the maximum R^2^ factor of the linear interpolation of the stress–strain curves for each sample (≥0.95).

### 2.7. Cell Viability

A preliminary evaluation of the cell viability at increasing concentrations of free DXM after 3 and 7 days of incubation was made. Thirty thousand normal human dermal fibroblast cells (HNDFs—P.5) were seeded on a 24-well plate and filled with 1 mL of DMEM 10% *v*/*v* culture medium (low glucose, GlutaMAX™ Supplement, pyruvate, supplemented with 10% FBS and 1% antibiotic) and 1 mL of each DXM concentration prepared in DMEM (0, 0.1, 0.05, 0.025, 0.0125, and 0.00625 mg/mL) to reach a final volume of 2 mL. Triplicates were made for each concentration. At fixed timing points, the cell culture medium was removed, and the samples were washed with 1 mL of PBS sterile solution. The PBS was removed and 1.5 mL of fresh PBS and 300 μL of MTT solution (5 mg/mL) were added to each sample. Samples treated with the MTT solution were incubated at 37 °C and 5% CO_2_ for 2 h and 30 min. After that time, the medium was removed and 1 mL of Dimethyl Sulfoxide (DMSO) was added for formazan salt solubilization. The samples were analyzed using a microplate reader at 570 nm (Microplate Photometer MPP-96, HiPo Biosan, Nebikon, Switzerland).

The placebo and DXM-loaded scaffolds (EL-S and HS) were kept under a vertical laminar flow safety hood overnight in order to promote the evaporation of organic solvent residues. Then, the scaffolds were cut (1.5 × 1.5 cm) and weighed to obtain samples bearing the same DXM concentration that was identified in the previous MTT test as giving the best results in terms of the cell viability %. The scaffolds were inserted into a CellCrown^TM^ System that reduced the scaffold floating during incubation. Scaffolds in the CellCrown^TM^ System were located onto a 12-well plate and left under UV overnight for sterilization. A suspension of 300,000 HNDFs was seeded on the surface of each scaffold and then filled with 2 mL of culture medium DMEM 10% *v*/*v* (low glucose, GlutaMAX™ Supplement, pyruvate, supplemented with 10% FBS and 1% antibiotic). The samples were left in an incubator at 37 °C and 5% CO_2_ for 7 days. Afterwards, the cell culture medium was removed, and the samples were washed with 2 mL of PBS sterile solution. The PBS was removed and 500 μL of MTT solution (5 mg/mL) in PBS was added to each sample; furthermore, 1.5 mL of PBS was added to cover all samples. The samples treated with the MTT solution were incubated at 37 °C and 5% CO_2_ for 2 h and 30 min. After that time, the medium was removed, and the samples were washed with PBS. Formazan salts and scaffold solubilization was obtained using 1 mL of THF. Negative Ctrls (Ctrl -) were made from scaffolds incubated in the absence of cells. The samples were analyzed using spectrophotometric analysis at 240 nm using quartz cuvettes (6705 UV/Vis Spectrophotometer, Single Mobile Holder, Jenway, Sungai Petani, Kedah, Malaysia).

### 2.8. Biological Characterization and Staining

SEM was used to investigate the cell morphology of the placebo and drug-loaded scaffolds. HNDFs (300,000 cells/scaffold) were seeded on the scaffold and 2 mL of culture medium was added. The medium was changed every 2 days and the cells were incubated for 7 days. After that time, the medium was removed, the scaffolds were washed with PBS, and then glutaraldehyde 0.4% was added and left for 10 min in order to fix the cells. After 10 min, the excess glutaraldehyde was aspirated and the scaffolds were washed twice with 1 mL of PBS so as to remove all glutaraldehyde traces. Dehydration was carried out with ethanol (EtOH) at increasing concentrations (30%, 70%, 80%, and 100%); each passage involved 10 min of contact with ethanol. Then, the samples were washed with a 50:50 mixture of dry EtOH 100% and hexamethyldisiloxane (HDMS) for 10 min. The samples were left to dry for 45 min to eliminate the solvents. The prepared samples were kept in a refrigerator until they were analyzed via SEM. 

Dapi (4′,6-diamidino-2-phenylindole, Thermofisher, Milano, Italy) and Phalloidin (Alexa Fluor™ 488 Phalloidin, Thermofisher, Milano, Italy) staining were performed on the HNDFs that were seeded on the scaffolds and incubated for 7 days. A total of 300,000 cells were seeded on the scaffold and 2 mL of culture medium was added. The medium was changed every 2 days and the cells were incubated for 7 days. After the medium was removed, the scaffolds were washed with PBS, and then glutaraldehyde 0.4% was added and left for 10 min in order to fix the cells. After 10 min, the excess glutaraldehyde was aspirated and washed twice with 1 mL of PBS so as to remove all glutaraldehyde traces. The cell membrane permeabilization was obtained through the addition of a solution composed of 0.1% Triton-X in PBS. After 10 min of incubation, the samples were washed again with PBS to remove the excess Triton-X, and were then treated with 50 μL of DAPI 300 nM solution and 50 μL of Phalloidin-Atto 488 solution for 3 h, protected by light. After staining, the samples were further washed with 1 mL of PBS to remove any Dapi/Phalloidin surplus, and mounted onto microscope glass slides. The samples were analyzed with a fluorescence microscope (Leica DM IL LED, Leica microsystems, Milano, Italy) and the resulting images were processed using ImageJ software. 

### 2.9. Statistical Analysis 

The reported graphs were plotted using Microsoft Excel. All experiments were carried out in triplicates, three times (n = 3), unless otherwise stated. All data are presented as the mean ± standard deviation (n = 3, unless stated otherwise). A statistical analysis tool was used in Microsoft Excel version:16.24 (Office 365 Microsoft, Redmond, WA, USA) using an analysis of variance (ANOVA). A *p*-value of less than 0.05 (*p* < 0.05) was considered statistically significant, and 0.05 < ** *p* ≤ 0.1 was marginally statistically significant, while a *p*-value higher than 0.1 (*p* > 0.1) was not statistically significant and indicated strong evidence for a null hypothesis. 

## 3. Results

### 3.1. Dexamethasone Quantification 

The amount of DXM loaded in the EL-S (#4), HD20-S (#5), and HD50-S (#6) was quantified and the results of the drug content (DC%) and encapsulation efficiency (EE%) are reported in Table 2. The weight of the fibers represents 20 ± 5% of the hybrid scaffolds’ (HD20-S-#5 and HD50-S-#6) weight, as explained previously [36].

The EE% of the EL-S (#4) was 80.9 ± 10.4%, and the EE% of the hybrid scaffolds with 20% (#5) and 50% (#6) infill were, respectively, 93.75 ± 1.5% and 85.0 ± 1.94%. This shows that the percentage of DXM contained in each scaffold reflects the DXM theoretic amount quite well, and is reproducible for all scaffolds obtained. The evaluated drug content confirms that the drug/polymer ratio of 0.2% *w*/*w* was maintained as a constant. 

### 3.2. Wettability

The contact angle (θ) of the placebo (#1, #2, and #3) and DXM-loaded (#4, #5, and #6) scaffolds was evaluated. The values at the starting point (0 min) and after an established contact time (2 min) with a drop of Hepes 0.1 M are shown in Table 3.

Considering the values of θ found for the placebo (#1) and DXM-loaded electrospun scaffolds (#4), it can be noted that after 2 min, the DXM-loaded electrospun scaffolds showed better wettability.

The contact angle values of scaffolds 3DP20-S and 3DP50-S showed lower values compared to the values obtained from both the placebo (#2 and #3) and DXM-loaded (#5 and #6) electrospun scaffolds. This demonstrates an apparent better wettability of the 3D-printed scaffolds, which can be explained due to the 3D-printed scaffolds having a particular infill structure that is highly porous compared to electrospun nanofibers, which have larger holes embedded in their structures, as is possible to see in Figure 1. A more detailed analysis of the pore size is assessed in the Section 3.4.

### 3.3. In Vitro Release Test in Static and Dynamic Conditions

In vitro release tests were conducted on the drug-loaded scaffolds for 7 days. The studies were carried out in Hepes 0.1 M (pH 7.2) in both static and dynamic conditions (at a flow rate of 0.4 mL/min) and samples were taken at the established times. Figure 2 reports the DXM release rate in Hepes 0.1 M at 37 °C. 

In all cases, the DXM release was prolonged by the fibers’ incorporation compared to the free DXM (black line) in the same conditions. The graph displays how the static conditions for scaffold #4 (grey line) show a significantly slower release rate in the first hours than those produced in dynamic conditions by the same sample (yellow line). In dynamic conditions, 58.55 ± 11.8% of DXM was released in the first 2 h, compared to 32.39 ± 2.18% of the drug being released in the same time frame in static conditions. As a result, it can be stated that dynamic release conditions promote the drug release rate in the first 4 h, but a complete release of DXM (100%) was achieved with the electrospun fibers in both cases in 24 h.

The DXM release from the two hybrid drug-loaded, 3D-printed, infill-electrospun scaffolds (samples #5 and #6, with 20% and 50% infill, respectively) was prolonged up to 48 h in both conditions tested (static and dynamic). In this case, no significant differences were observed between the static and dynamic release profiles. Comparing the DXM release profiles between the EL-S-DXM and the HD-S-DXM, it is conceivable that the DXM release from the electrospun fibers (EL-S-DXM) was faster because they do not have the one-side constraint of the 3D-printed structure, which slows down the drug diffusion.

The drug release kinetics of DXM from the EL-S-DXM, HD20-S-DXM, and HD50-S-DXM in static (S.C.) and dynamic conditions (D.C.) were evaluated. The results are shown in Table 4. 

Considering the R^2^ values, the DXM in vitro release behavior results under static conditions followed first-order kinetics, where the release rate was directly proportional to the drug concentration, whereas the release profiles in dynamic conditions for scaffolds #4 and #6 followed Higughi kinetics. The Higuchi model expresses the drug release from a system that involves both dissolution and diffusion mechanisms. If it is confirmed that the drug release diffusion is controlled via the Higuchi equation, it is possible to plot the dissolution data using the Korsmeyer–Peppas model. The n value for the EL-S-DXM (sample #4) is 0.1194, indicating an almost Fickian diffusion, and the *n* value for the HD50-S-DXM (sample #6) is 0.6803, which describes a non-Fickian transport, which suggests first-order kinetics. In fact, the value of R^2^ in first-order kinetics for scaffold #6 is very high (0.9335). The same behavior is highlighted for the DXM release kinetic of scaffold #5. Therefore, it can be stated that the DXM release kinetics of electrospun fibers follow a Fickian diffusion model, and the addition of a 3D-printed polymer support leads to changes in the drug release kinetics. 

### 3.4. Morphologic Characterization

A preliminary morphologic characterization was performed via SEM on the electrospun and hybrid scaffolds, both placebo and drug-loaded (Figure 3). The EL-S and EL-S-DXM scaffolds were characterized by the presence of homogeneous nanofibers, forming an interconnected network with a high surface area and porosity. SEM images and a morphological analysis of the plain 3D-printed (20% and 50% infill) scaffolds were performed in a previous work by the authors, and showed polymeric filaments with a full, smooth, and non-porous structure; the porosity of the scaffold was given based on the distance between the deposited filaments [36].

The hybrid scaffolds showed a complete coverage and adhesion between the 3D scaffolds at different infills of 20% and 50% and the electrospun fiber layer (Figure 3b,c). Moreover, it was observed that the electrospun fibers maintained nanometric and homogeneous dimensions, despite being electrospun on an inert 3D-printed polymer support instead of on a conductive metal collector. In fact, the presence of a 3D polymeric structure on the collector did not interfere with the electrospinning and fiber deposition process. This is because the porosity and thickness of the 3D polymeric scaffolds allowed the conductivity of the collector to be maintained.

The drug-loaded fibers did not show notable morphological differences compared to the placebo ones (Figure 3).

Table 5 reports the results of the electrospun fibers’ diameter, surface porosity, and pore surface area, obtained through ImageJ analysis.

The surface porosity % showed values of about 30% for all the analyzed samples, showing that they are suitable for guaranteeing fibroblast cell adhesion and proliferation, as reported in the literature [8]. 

Further characterization was performed to evaluate the possible differences that may have arisen after the in vitro DXM release from sample #4, either in static or dynamic conditions. The images obtained via SEM (Figure 4) were processed with ImageJ, and data on the fibers’ diameter (μm ± SD), pore area (μm^2^ ± SD), and porosity % were generated, and are reported in Table 6. 

The obtained data and images show that the fibers increased in size after having released the drug. This occurred under both static and dynamic conditions. In addition, there was also an increase in the porosity and pore size due to the fibers swelling after the drug release. The highest correlation coefficient ($R^{2}$) derived from the Hixson–Crowell kinetics model (shown in Table 4) can be correlated to this behavior, proving that the release process involves a change in the surface area and diameter of the drug-loaded fibers. 

### 3.5. Comparative Mechanical Assessment

The results of the mechanical characterization that was performed on the placebo and drug-loaded samples are reported in Table 7. The E reference values from the previous mechanical characterization of the placebo scaffolds are also reported. 

The reported E (MPa) values were evaluated using the maximum R^2^ factor that was calculated in the initial portion of the stress–strain curve, i.e., the strain % range listed for each sample in Table 7. The E values that were evaluated for all placebo scaffolds are in line with the E values obtained from the previous characterization [36], proving the scaffolds’ manufacturing reproducibility. The Young’s modulus (#1, E = 2.2 ± 0.3) of the placebo electrospun fibers was slightly lower compared to the E value of the DXM-loaded electrospun fibers (#4, E = 6.8 ± 1.8). With regard to the 20% infill hybrid scaffolds, the presence of the fibers with the drug (#5) seemed to increase the E value and, therefore, the mechanical resistance. Instead, for the hybrid scaffolds with 50% infill, there were no substantial E value differences that exceeded 40 MPa, and so they are not very usable in the field of soft tissue regeneration [44].

### 3.6. Biological Characterization and Staining

To assess the cell viability of fibroblast growth at increasing concentrations of dexamethasone (DXM), an MTT assessment was made at 3 and 7 days. The obtained results are reported in Figure 5.

It is stated in the literature that DXM has effects on proliferation, depending on the dosage. After the first three days of incubation with DXM, there was a much greater growth than the control. A DXM concentration of 0.0125 mg/mL gave greater results in terms of cell viability. However, after 7 days in culture, the vitality tends to decrease, returning to the values similar to those reported for the control. Most likely, the high cell growth led to the premature attainment of a state of confluence of the cells in the well with an increased production of cell waste products, which caused the premature death of the cells.

The cell viability of fibroblast growth on the scaffolds was evaluated after 7 days using an MTT assay. The obtained results are reported in Figure 6.

SEM characterization was performed on the cellularized scaffolds after a 7-day incubation period (Figure 7).

The obtained images show the presence of adherent cells on the electrospun scaffold surface. The cells’ penetration into the electrospun nanofibrous matrices of both the placebo and DXM fibers is confirmed. Moreover, as reported in the MTT test, the SEM analysis confirmed that there were more cells on the DXM-loaded scaffolds than on the placebo ones.

At the same point in time (7 days), Dapi/Phalloidin staining analysis was also performed (Figure 8).

The Dapi/Phalloidin staining provides further confirmation of the presence of cells adhered to the surface of the scaffolds. In the DXM-loaded scaffolds, the adhesion and proliferation seem more evident, in line with the results obtained from the MTT test and SEM characterization. 

## 4. Discussion

In this article, we wanted to investigate, as a proof of concept, the possibility of using electrospun and hybrid scaffolds based on PLA/PCL as a platform for controlled drug release and the concomitant support of cell growth. The selected copolymer, PLA/PCL, is notoriously biocompatible and biodegradable; however, it can be seen how different manufacturing techniques (electrospinning or 3D-printing) modifies the structural and biological properties of the obtainable scaffold. Also, from the point of view of the drug release, the presence of a 3D support creates a physical “obstacle” that is capable of slowing down the release compared to the electrospun scaffolds alone. However, the decision to develop a hybrid scaffold stems from the need to make electrospun scaffolds easier to handle and more resistant, and, at the same time, to make 3D-printed scaffolds suitable as a support for adhesion and cell growth, even at very low infills (20% and 50%). In fact, the same authors have demonstrated in a previous article how seeding that was performed only on PLA/PCL 3D scaffolds that were printed at different infills was not very functional, as the cells tended to pass through the porosity and adhere to the bottom of the multiwell rather than adhere to the scaffolds. The addition of an electrospun layer over the 3D support has made it possible to overcome this problem, and has also created a higher-performing system in terms of mechanical strength and cell viability compared to the electrospun scaffold alone [36]. Furthermore, with the FFF technique for the production of 3D scaffolds, the loading of a drug becomes difficult, as the high temperatures that are used can cause the degradation of the active ingredients and/or crystallinity changes. On the contrary, the electrospinning technique does not exploit high temperatures, so there is no risk of damage to the active substance. By assembling a 3D support and DXM drug-loaded fibers into the hybrid scaffold, a controlled-release drug delivery system can be obtained.

## 5. Conclusions

The hybrid scaffolds that were produced by combining the 3D-printing technique with electrospinning have demonstrated how the integration of the two techniques is functional to obtain scaffolds with a defined macroscopic geometry, but that are also characterized by the presence of polymeric nanofibers that are able to better mimic the nanoarchitecture of the native ECM for promoting soft tissue regeneration. From the results obtained, it was seen how the presence of polymeric nanofibers on the 3D scaffolds favored cell adhesion and growth. Furthermore, the versatility of the electrospun fibers allows them to be used as a drug delivery platform in order to favor the regenerative process. In this case, in terms of which method of DXM delivery was used, it was seen that the release (prolonged up to 48 h with the hybrid scaffolds) and the kinetics allowed for a superior cell growth and proliferation (>200%) to be obtained compared to the placebo scaffolds.

In summary, the hybrid scaffolds that were developed both as a placebo and as a drug carrier can be optimized and developed as a versatile platform according to specific mechanical needs (organ/tissue target) and the type of drug (anti-inflammatory or antibiotic) treatment required.

## Figures and Tables

**Figure 1 pharmaceutics-15-02478-f001:**
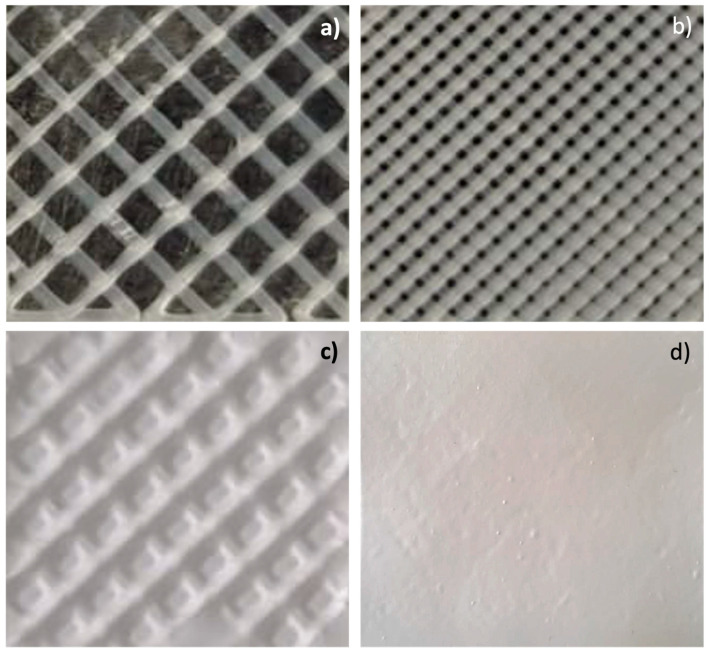
Images of (**a**) 3D-printed scaffold with 20% infill (3DP20-S); (**b**) 3D-printed scaffold with 50% infill (3DP50-S); (**c**) placebo hybrid scaffold (HD20-S); and (**d**) placebo hybrid scaffold (HD50-S).

**Figure 2 pharmaceutics-15-02478-f002:**
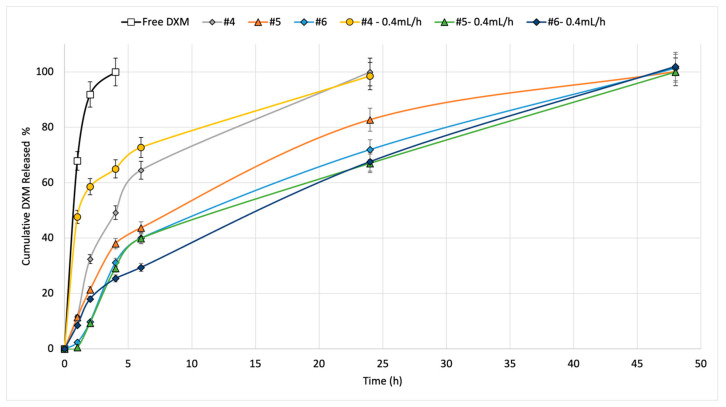
In vitro release of DXM at 37 °C in Hepes 0.1 from electrospun fibers in static (grey line) and dynamic (yellow line) conditions; HS 20% infill in static (orange line) and dynamic (green line) conditions; HS 50% infill in static (light blue line) and dynamic (blue line) conditions.

**Figure 3 pharmaceutics-15-02478-f003:**
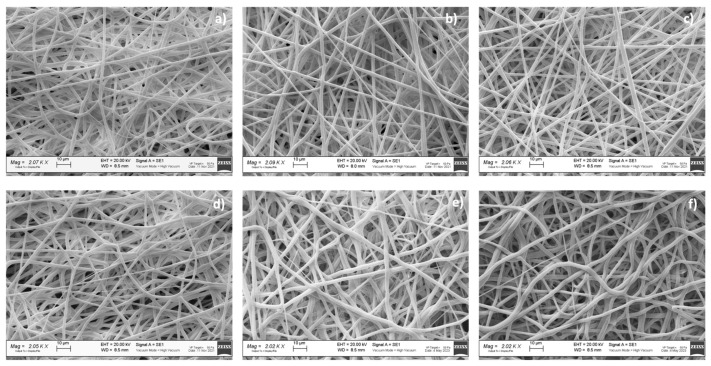
SEM images of (**a**) EL-S (#1); (**b**) HD20-S (#2); (**c**) HD50-S (#3); (**d**) EL-S-DXM (#4); (**e**) HD20-S-DXM (#5); and (**f**) HD50-S-DXM (#6).

**Figure 4 pharmaceutics-15-02478-f004:**
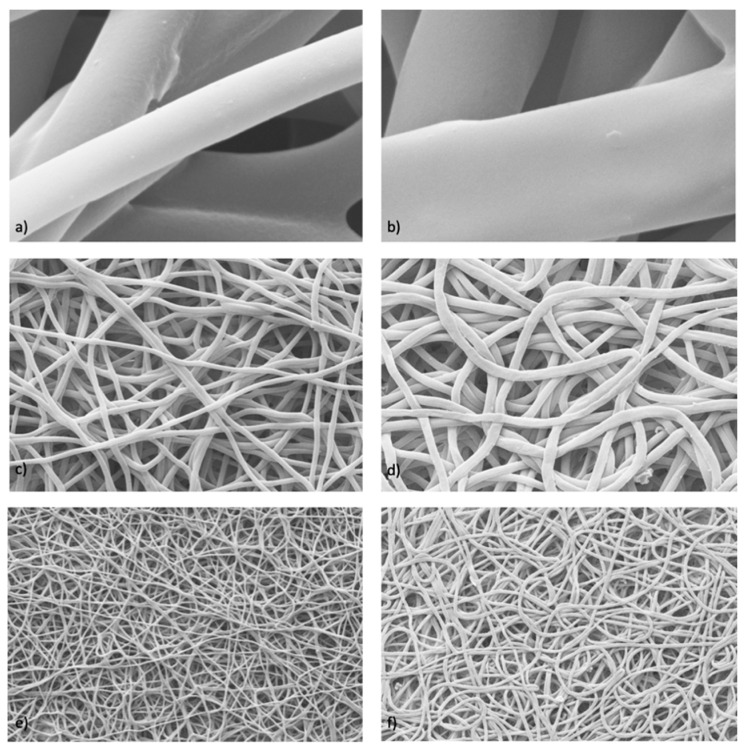
SEM analysis using ImageJ of electrospun fibers with DXM (#4) obtained after an in vitro release test in static (**a**,**c**,**e**) and dynamic (**b**,**d**,**f**) conditions.

**Figure 5 pharmaceutics-15-02478-f005:**
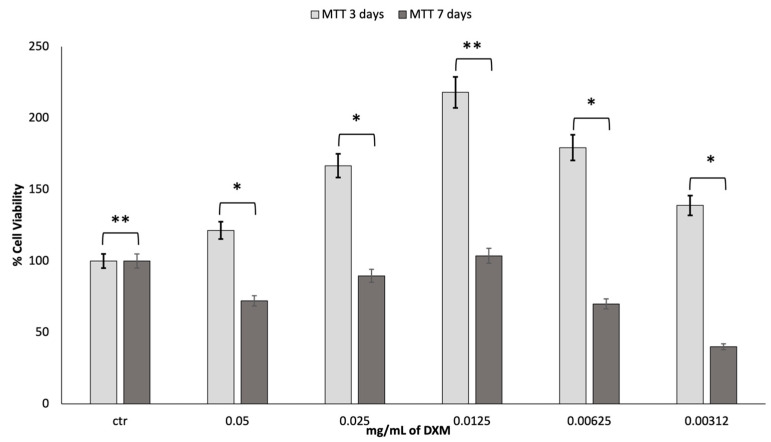
Results of MTT test after 3 and 7 days. * *p* ≤ 0.05 is statistically significant; ** *p* ≥ 0.05 is not statistically significant.

**Figure 6 pharmaceutics-15-02478-f006:**
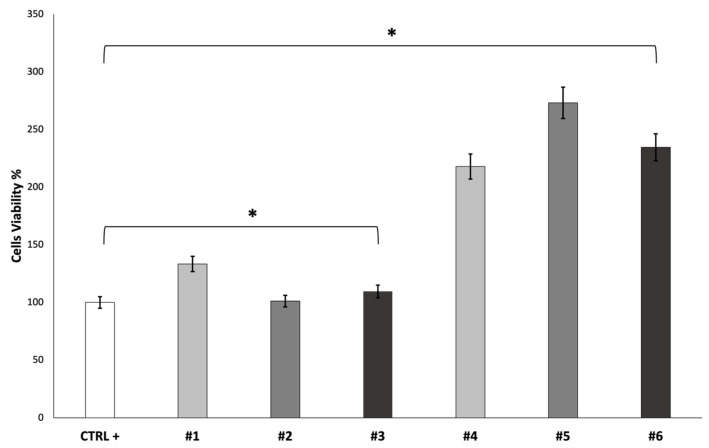
Results of MTT test after 7 days. * *p* ≤ 0.05 is statistically significant.

**Figure 7 pharmaceutics-15-02478-f007:**
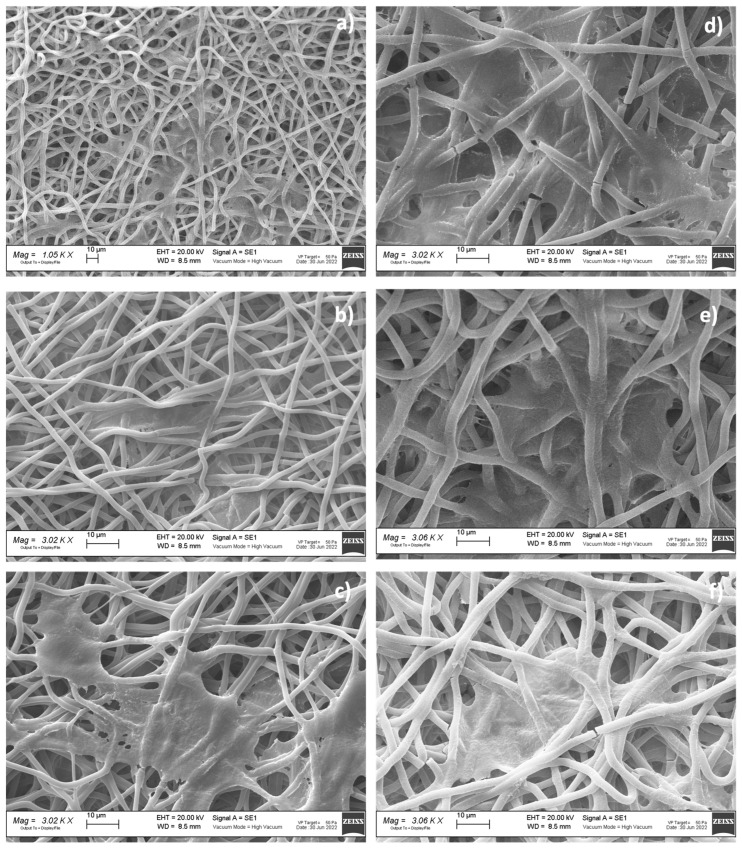
SEM images performed 7 days after HNDF incubation on (**a**) EL-S (#1); (**b**) EL-S, DXM-loaded (#4); (**c**) HD20-S (#2); (**d**) HD20-S, DXM-loaded (#5); (**e**) HD50-S (#3); and (**f**) HD50-S, DXM-loaded (#6).

**Figure 8 pharmaceutics-15-02478-f008:**
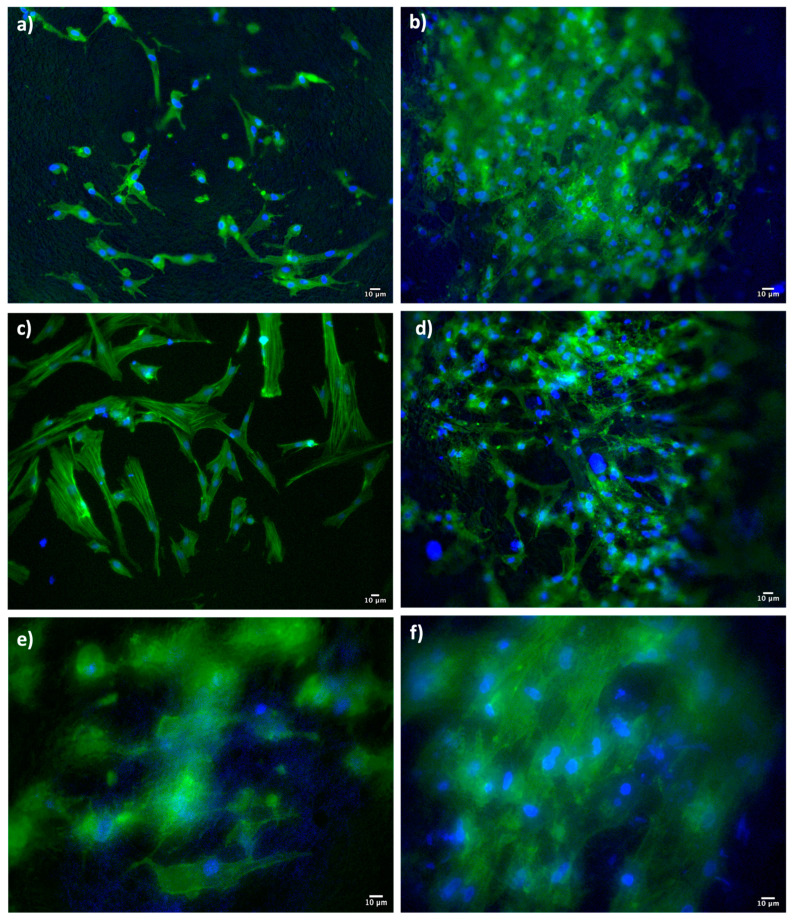
Dapi/Phalloidin staining performed 7 days after the HNDF seeding of (**a**) EL-S (#1); (**b**) EL-S, DXM-loaded (#4); (**c**) HD20-S (#2); (**d**) HD20-S, DXM-loaded (#5); (**e**) HD50-S (#3); and (**f**) HD50-S, DXM-loaded (#6).

**Table 1 pharmaceutics-15-02478-t001:** Placebo and DXM-loaded PLA/PCL scaffolds.

Sample	Acronym	Scaffold	DXM
#1	EL-S	Placebo electrospun scaffold	-
#2	HD20-S	Hybrid placebo 3D-printed 20% infill electrospun scaffold	-
#3	HD50-S	Hybrid placebo 3D-printed 50% infill electrospun scaffold	-
#4	EL-S-DXM	Drug-loaded electrospun scaffold	0.04% *w*/*v*
#5	HD20-S-DXM	Hybrid drug-loaded 3D-printed 20% infill electrospun scaffold	0.04% *w*/*v*
#6	HD50-S-DXM	Hybrid drug-loaded 3D-printed 50% infill electrospun scaffold	0.04% *w*/*v*

**Table 2 pharmaceutics-15-02478-t002:** Drug content % and encapsulation efficiency % results.

Scaffold Type	Scaffold (1.5 × 1.5 cm) Weight (mg)	Fiber Weight (mg)	DXM Theoretic Amount (mg)	DXM Experimental Amount (mg)	DC%	EE%
#4	12.6 ± 0.56	12.6 ± 0.56	0.02 ± 0.006	0.020 ± 0.002	0.16 ± 0.02	80.9 ± 10.4
#5	80.96 ±22.58	16.19 ± 4.5	0.032 ± 0.003	0.030 ± 0.006	0.20 ± 0.003	93.75 ± 1.5
#6	85.93 ±13.40	17.18 ± 2.7	0.033 ± 0.005	0.028± 0.005	0.16 ± 0.003	85.0 ± 1.94

**Table 3 pharmaceutics-15-02478-t003:** Values of contact angle (θ) (for the sake of clarity, the description of the scaffold composition has been reported from Table 1).

Sample	Scaffold	*θ* (0 min)	*θ* (2 min)
#1	EL-S	99.6 ± 0.29	97.13 ± 0.23
#2 (electrospun side)	HD20-S	101.0 ± 1.15	97.33 ± 0.23
#3 (electrospun side)	HD50-S	97.43 ± 0.15	93.86 ± 0.42
#4	EL-S-DXM	91.27 ± 1.98	90.76 ± 0.15
#5 (electrospun side)	HD20-S-DXM	99.0 ± 0.26	96.8 ± 0.10
#6 (electrospun side)	HD50-S-DXM	93.8 ± 0.53	92.73 ± 1.03
3D-printed scaffold 20% infill	3DP20-S	82.27 ± 0.63	79.03 ± 0.23
3D-printed scaffold 50% infill	3DP50-S	84.73 ± 0.81	82.43 ± 0.57

**Table 4 pharmaceutics-15-02478-t004:** Release kinetics of DXM from EL-S-DXM (#4), HD20-S-DXM (#5), and HD50-S-DXM (#6) in static (S.C.) and dynamic conditions (D.C.).

Release Kinetics	Zero-Order	First-Order		Higuchi	Korsmeyer–Peppas		Hixson–Crowell
	*R* ^2^	*R* ^2^	*K* _1_	*R* ^2^	*R* ^2^	*n*	*R* ^2^
Static conditions							
#4	0.9667	0.9935	−0.0756	0.9488	0.9496	0.9283	0.9889
#5	0.9519	0.9733	−0.0428	0.9656	0.9824	0.7719	0.9672
#6	0.9693	0.9704	−0.0404	0.8396	0.9745	1.6252	0.9704
Dynamic conditions							
#4	0.6698	0.6698	−9.7808	0.908	0.9532	0.1194	0.8723
#5	0.9705	0.9710	−0.0402	0.8210	0.9117	2.4154	0.9714
#6	0.9107	0.9335	−0.0249	0.9768	0.9456	0.6803	0.9263

**Table 5 pharmaceutics-15-02478-t005:** Fibers’ diameter, surface porosity, pore surface area, and thickness for placebo and drug-loaded samples.

Sample	Electrospun Fiber Diameter (nm)	Surface Porosity (%)	Pore Surface Area (mm^2^)	Thickness (mm)
#1	732 ± 28	33.94 ± 1.12	0.0047 ± 0.0021	0.15 ± 0.04
#2	766 ± 41	34.14 ± 1.06	0.085 ± 0.1	0.515 ± 0.04
#3	755 ± 37	36.92 ± 1.02	0.069 ± 0.04	0.730 ± 0.04
#4	773 ± 57	35.88 ± 2.42	0.012 ± 0.009	0.17 ± 0.2
#5	795 ± 22	35.54 ± 1.45	0.075 ± 0.5	0.550 ± 0.04
#6	789 ± 32	37.32 ± 2.42	0.089 ± 0.03	0.655 ± 0.04

**Table 6 pharmaceutics-15-02478-t006:** Sample #4: results of the fibers’ diameter (μm± SD), pore area ($\mu m^{2}\pm$ SD), and porosity % after release in static and dynamic conditions.

#4	Static Conditions	Dynamic Conditions
Fiber Diameter (μm)	1.5 ± 0.1	2.9 ± 0.3
$\mathrm{Pore}\;\mathrm{Area}\;(\mu m^{2}$)	0.012 ± 0.01	0.017 ± 0.02
Porosity %	44.4 ± 2.03	40.9 ± 4.40

**Table 7 pharmaceutics-15-02478-t007:** Results of Young’s modulus (E), 5 mm/min.

Sample	E (MPa)	Strain %	E Reference Values [37]
#1	2.2 ± 0.3	0 − 20 ± 8.7%	6.21 ± 3.45 MPa
#2	10.8 ± 1.4	0 − 3.2 ± 0.6%	7.39 ± 3.85 MPa
#3	46.4 ± 4.1	0 − 4.2 ± 2.5%	34.77 ± 6.37 MPa
#4	6.8 ± 1.8	0 − 14 ± 9.6%	-
#5	24.1 ± 1.5	0 − 3.75 ± 0.3%	-
#6	40.4 ± 5.8	0 − 3.25 ± 0.4%	-

## Data Availability

All data are contained in the article.

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
