# Peer review of "Hybrid 3D-Printed and Electrospun Scaffolds Loaded with Dexamethasone for Soft Tissue Applications"

_pharmaceutics, 2023, doi:10.3390/pharmaceutics15102478_

Round 1
Reviewer 1 Report
The nano soft tissue applications are important factor to the regenerative process after injuries. The electrospun nanofibrous tissue is a good way for the effective and efficient regeneration. There are some works can be done to improve the manuscript.
1, the nanofibrous morphology is the key to the tissue, how to control the nanofibrous morphology should be discussed more detail. What the morphology suitable for the regenerative application.
2, the effect of process parameters on the nanofibrous morphology.
3, the regenerative experiment should done.
English expression should be improved.
Author Response
Reviewer question: the nanofibrous morphology is the key to the tissue, how to control the nanofibrous morphology should be discussed more detail. What the morphology suitable for the regenerative application.
Author’s answer: We thank the reviewer for his comment. We agree that nanofibers morphology is a very important factor for tissue regeneration. Each tissue has particular structure, morphology, porosity and mechanical characteristics for which the polymeric nanofibers obtained must comply in order to mimic the original functionality. At production level through electrospinning technique, the parameters that influence the final fibers morphology are many such as, polymer concentration, syringe needle size, type of collector, temperature and humidity during the process. As mentioned before, each tissue has specific morphological characteristics and therefore the needs that polymeric nanofibers must satisfy are different, for this reason there is not a single morphology suitable for the regeneration of all tissues, but morphology must be adapted according to the target tissue [1,2]. As example, tendons, heart tissue, and blood vessels, exhibit anisotropic arrangements with highly ordered structures, while random fibers can be used for tissues without specific orientation, such as skin or adipose tissue. However, random fiber orientation is advantageous for tissue engineering as it offers increased surface area that promotes nutrient exchange, cell adhesion and proliferation, and as consequence it improves tissue regeneration. When pore sizes are large enough, randomly orientated fibers allow cell infiltration throughout the scaffold which helps with nutrient and oxygen exchange and waste removal[3,4].
At cellular level, cells morphology is influenced by fibers morphology; cells exhibit less elongated morphology when they grow on randomly oriented nanofibers, and low-porosity scaffolds, while cells exhibit elongated, spindle-shaped morphology when they grow on aligned nanofibers, and high-porosity scaffolds. Cells migrate with higher velocities on aligned nanofibers, and highly-porous scaffolds, but penetrate more on microfibers, random fibers, and highly porous scaffolds[5,6].
In the manuscript the part of how to control the nanofibrous morphology and applications in TE has been expanded in the Introduction.
Reviewer question: the effect of process parameters on the nanofibrous morphology.
Author’s answer: Electrospinning is a versatile method for the nanofibers producion from various kinds of polymers. The diameter and morphology of electrospun nanofibers are mainly governed by electrospinning parameters and play a key role in their final applications. Electrospinning parameters can be categorized into three main groups: (1) polymer solution (eg, concentration, viscosity, surface tension, and conductivity), (2) process (eg, applied electrostatic potential, collection dis- tance, and feed rate), and (3) ambient (eg, temperature, relative humidity (RH), and surrounding air velocity in the spinning chamber) parameters.
Concerning process parameters, several studies found that higher voltages caused a reduction on fiber diameter. Instead, the pore size and diameter of fibers increased when the flow rate increased. The type of collectors exerts significant influence on the electrospun fiber alignment. The conductive metal plate is commonly used as a target to orient the fibers randomly in a nonwoven structure; while using a cylindrical drum highly aligned fibers can be obtained[7,8].
Reviewer question: the regenerative experiment should done.
Author’s answer: We agree with the reviewer that in vitro and in vivo studies are needed to test the regenerative capacities. We are working on this by realizing hybrid scaffolds cellularised with mesenchymal cells on which we are inducing myogenic differentiation and then testing their activity in vitro and in vivo. The data we are obtaining will be the subject of a future publication.

Reviewer 2 Report
The aim of this study was to investigate the possibility of using electrospun and hybrid scaffolds based on Poly-L-lactide- co – poly- caprolactone (PLA-PCL) as a platform for controlled drug release and the concomitant support of cell growth. Dexamethasone (DXM) was loaded as anti-inflammatory drug, into electrospun fibers, and the drug release kinetics and scaffold biological behaviour were investigated. The fibers maintain nanometric dimensions even after deposition on 3D printed supports. Cell adhesion and proliferation is favored in DXM-loading hybrid scaffolds. The hybrid scaffolds developed can be enhanced as adaptable platform for soft tissue regeneration application.
The paper is interesting, but the authors need to address the following issues before the acceptance of this paper.
Minor issues:
1. 1. Although authors in their previous publication “Design of copolymer PLA-PCL electrospun matrix for biomedical applications” investigated the parameters which influence the properties of PLA-PCL electrospun fibers, they need in brief in this publication for the readers to address the following:
a. How influence of the distance between needle and target has on the morphology and nanofiber dimension?
b. The charge of the deposited fibers has influence of the morphology of the fibers, but there are other parameters such as solvent, which also contribute. The authors should make some comments related to the methylene chloride (MC) and dimethylformamide (DMF) factor on morphology of electrospun fibers, and why the MC:DMF 70:30 ratio was used in this study.
2. 2. In Conclusions section, summary of some results need to be addressed.
Author Response
The paper is interesting, but the authors need to address the following issues before the acceptance of this paper.
Minor issues:
1. 1. Although authors in their previous publication “Design of copolymer PLA-PCL electrospun matrix for biomedical applications” investigated the parameters which influence the properties of PLA-PCL electrospun fibers, they need in brief in this publication for the readers to address the following:
Reviewer question: How influence of the distance between needle and target has on the morphology and nanofiber dimension?
Author’s answer: Following the reviewer suggestion, the authors added more details on the influence of electrospinning process parameters. In example, itwas observed that the distance between the collector and the needle tip can also affect the final fiber diameter and morphology. If the distance is too short, the solvent cannot fully evaporate before it reaches to collector, on the contrary, a too-long distances caused the formation of beaded fibers. An optimum distance should be set up to allow the collection of dry electrospun nanofibers. Therefore, considering the polymeric concentration and the solvents optimized in previous works, the needle-collector distance of 15 cm proved to be the most suitable for obtaining dry fibers with dimensions below the micrometer. From this work it was also possible to observe how the presence of a 3D polymeric structure on the collector did not interfere with the electrospinning and fiber deposition process. However, this is because the porosity and thickness of the 3D polymeric scaffolds have maintained the conductivity of the collector.
Reviewer question: The charge of the deposited fibers has influence of the morphology of the fibers, but there are other parameters such as solvent, which also contribute. The authors should make some comments related to the methylene chloride (MC) and dimethylformamide (DMF) factor on morphology of electrospun fibers, and why the MC:DMF 70:30 ratio was used in this study.
Author’s answer: As reported in a previous study of the same authors and in the literatuee, MC was chosen because it is a good solvent for the PLA-PCL copolymer, and for its low boiling point (40 °C) that allows fast solvent evaporation during electrospinning process. DMF was selected for its high dielectric constant (36.70) that promotes fibers stretching when an electric field is applied. The 70:30 combination was derived from optimization studies where various blends ratio were evaluated [9].
Reviewer question: 2. In Conclusions section, summary of some results need to be addressed.
Author’s answer: We thank the reviewer for the suggestion. A summary of the results has been added in the conclusions.

Round 2
Reviewer 1 Report
This manuscript is suitable for publication.